# A Lightweight Leddar Optical Fusion Scanning System (FSS) for Canopy Foliage Monitoring

**DOI:** 10.3390/s19183943

**Published:** 2019-09-12

**Authors:** Zhouxin Xi, Christopher Hopkinson, Stewart B. Rood, Celeste Barnes, Fang Xu, David Pearce, Emily Jones

**Affiliations:** Department of Geography, University of Lethbridge, Lethbridge, AB T1K 3M4, Canada

**Keywords:** LiDAR, Leddar, sensor fusion, structure from motion, monocular camera, terrestrial laser scanning, leaf area index, canopy monitoring

## Abstract

A growing need for sampling environmental spaces in high detail is driving the rapid development of non-destructive three-dimensional (3D) sensing technologies. LiDAR sensors, capable of precise 3D measurement at various scales from indoor to landscape, still lack affordable and portable products for broad-scale and multi-temporal monitoring. This study aims to configure a compact and low-cost 3D fusion scanning system (FSS) with a multi-segment Leddar (light emitting diode detection and ranging, LeddarTech), a monocular camera, and rotational robotics to recover hemispherical, colored point clouds. This includes an entire framework of calibration and fusion algorithms utilizing Leddar depth measurements and image parallax information. The FSS was applied to scan a cottonwood (*Populus* spp.) stand repeatedly during autumnal leaf drop. Results show that the calibration error based on bundle adjustment is between 1 and 3 pixels. The FSS scans exhibit a similar canopy volume profile to the benchmarking terrestrial laser scans, with an *r*^2^ between 0.5 and 0.7 in varying stages of leaf cover. The 3D point distribution information from FSS also provides a valuable correction factor for the leaf area index (LAI) estimation. The consistency of corrected LAI measurement demonstrates the practical value of deploying FSS for canopy foliage monitoring.

## 1. Introduction

Monitoring in-situ canopy variables, such as leaf area index (LAI) from the ground is valuable for developing canopy light interception and biomass growth models for forests [1,2]. Passive optical sensors, such as digital cameras, are cost-effective ground-based tools, and have been applied to monitor canopy variables, such as LAI, and the fraction of absorbed, photosynthetically active radiation (fPAR) [3]. However, the passive optical approach provides limited accuracy due to the 3D heterogeneity of foliage distribution. For example, the LAI estimated from the Beer–Lambert geometric-optical model [4,5], also termed effective LAI, is usually 55%–65% of the true LAI [6]. The constraint of sensing tools inevitably leads to sophisticated tuning efforts for the geometric-optical model, such as the introduction of gap size distribution, clumping factor, and needle-to-shoot area ratio [7]. The emerging terrestrial laser scanning (TLS) technology significantly mitigates the LAI characterization problems. The 3D datasets from TLS not only enable straightforward estimation of canopy model variables, including leaf angle distribution (LAD) [8,9], clumping index [10], canopy foliage profile [11], gap fraction [12], and plant area volume density (PAVD) [13,14], but have also led to the development of more accurate canopy models. For example, the non-randomness of leaf distribution, conventionally described using the clumping index or gap size distribution, can be explicitly modeled in path length distribution equations [15,16]. Basically, the path length distribution (PATH) model relates the light extinction degree with the detailed optical paths extractable from 3D point clouds. It is adapted from the foliage profile equation in [17]. The LAI variable can be estimated from the PATH model with high accuracy and stability benchmarked by the true LAI [7,15]. Therefore, capturing 3D information has substantial potential for modeling canopy variables.

Conventional TLS sensors have shown impressive measurement precision and reliability of capturing 3D canopy information, yet lack portability and affordability for widespread use at the landscape level. An alternative solution is to integrate low-cost LiDAR sensors into an existing ground sensor network for broad-scale monitoring purposes. Manufacturers such as Faro and Leica produce portable terrestrial or mobile LiDAR at a price range of 4000 to 20,000 USD with moderate frequency (20–300 s^−1^) and centimeter-level resolution. Those sensors are not sufficiently cost effective, power saving, compact, or flexible for widespread 3D biomass monitoring. In recent years, tiny LiDAR scanners, such as those available from Velodyne, Ouster, Hokuyo, SICK, Ibeo, and Scanse have entered the market with a price level between 100 and 4000 USD. Most tiny scanners have a limited field of view (FOV) and low point detection frequency. Many low-cost tiny scanners scan in only 2D, recording laser distance returns with a spinning mirror or motor. Thus, multiple scanlines are required to produce sufficient vertical points, whereas low frequencies can cause serious 3D distortions from fast-moving platforms or targets [18].

A multi-segment (sometimes referred to as multi-beam) laser scanner may provide a balanced choice of budget and frequency. Instead of repetitive scanning in the vertical direction, a multi-segment scanner relies on detection arrays to record multiple distances instantly. Each segment from a detection array usually has high detection frequency over 50 s^−1^. Among the multi-segment scanners, an LED-based LiDAR from LeddarTech, known as Leddar (light emitting diode detection and ranging), stands out as an economical choice, with 16 segments, 100 s^−1^ frequency, and a thousand-dollar cost. A Leddar sensor implements patented algorithms to estimate traveling distances of each pulse emitted from an LED light source and detected by an array of 16 PIN photodiodes [19]. Each segment corresponds to a solid angle of approximately 2.8° × 7.5° and the field of view for 16 segments is customizable between 9° and 95°. The sensor can record multiple distance returns from multiple objects at different distances. Leddar’s capability of rapid data acquisition and multiple object detection has enabled growing applications in canopy detection [20], autonomous driving [21,22], traffic analysis [23,24], parking assistance [25], and drone altitude estimation [26,27]. However, due to limited FOV and sparse segments, the Leddar sensor alone cannot compete with conventional static TLS for detailed 3D canopy modeling or geometric mensuration.

Adding a twin-axis rotational robot to the Leddar sensor can be a cost-effective solution to expand its FOV and enhance point clouds. The boresight of the integrated system, however, needs to be calibrated in order to deliver precise and consistent 3D data or point clouds. One type of LiDAR calibration method is statistical correction based on a known target. For example, Bohren et al. [28] correct 3D ground point clouds from SICK and Velodyne scanners based on the planarity constraint of the ground. A more rigorous calibration method is to model the physical relationship between the LiDAR system and calibration targets. For example, Muhammad and Lacroix [29] use a planar target to calibrate five intrinsic parameters of a Velodyne HDL-64E S2 system mounted on a static rotator, including two segment angles and three origin parameters. Atanacio-Jiménez et al. [30] expand the calibration target to be a room of five planes, and calibrate intrinsic and extrinsic parameters of a Velodye HDL-64E. Zhu and Liu [18] estimate the origin and orientation of the HDL64E S2 sensor by aligning point clouds based on pole-shaped features. Several other studies provide convenient physical calibration approaches without a requirement for measuring reference target coordinates. For example, Levinson and Thrun [31] propose a global calibration method for 192 orientation or distance parameters generated from a moving trajectory of a Velodyne HD-64E S2 sensor. Point clouds are self-calibrated by maximizing local planarity without a need for a deliberate reference target. Similarly, Sheehan et al. [32] proposed an entropy-based self-calibration method, maximizing the sharpness of point clouds for three SICK LMS-151 laser scanning units.

The above calibration methods require accurate position and orientation data from external sources, such as a wheel encoder, GPS, or the inertial measurement unit (IMU), and also dense points to support point cloud alignment and registration. These requirements are usually not easily met in a low-cost scanning system with Leddar and a rotational motor. An alternative solution is to integrate a camera sensor. The extrinsic parameters, such as pose and origin, can be estimated using photogrammetric methods. The predominant approach from existing studies is to re-project LiDAR point clouds to a 2D plane and co-register with an image based on corresponding features [33,34,35,36,37,38]. The calibration accuracy from these studies substantially depends on highly dense point clouds, which a low-cost Leddar sensor is unable to produce. Among the few studies on calibrating sparse LiDAR data, Debattisti et al. [39] places focus on edge points of artificial targets visible from both an image and a SICK LMS221 sensor. Debattisti, Mazzei, and Panciroli [39] also point out that point clouds from low-cost LiDAR usually lead to laborious scanning in pursuit of sufficient corresponding points from both the image and point cloud. Considering the characteristics of a low-cost scanning system, this study will utilize a planar calibration target to avoid the prerequisite for dense point clouds and also integrate a camera to provide extrinsic pose estimation.

Adding a camera sensor to the scanning system not only satisfies the calibration and alignment need but also provides useful texture details. A question of interest is how to integrate the texture information from a camera with the sparse point clouds from a Leddar scanning system, in order to produce densely colored point clouds. Indeed, a variety of existing literature, including Hartley and Zisserman [40] in particular, already illustrates the feasibility of reconstructing dense 3D point clouds from stereo or multiple images directly, without LiDAR distance. However, in regard to our small scanning system with one rotational camera, the short movement baseline would lead to poor 3D reconstruction quality. Assuming the target is far from the camera, the error of depth estimate (ΔD) is related to the error of stereo matching between two images (Δx) in Equation (1):(1)$$\Delta x\;=\;\frac{fd\Delta D}{\mu D^{2}}$$

Assuming the baseline d of the small scanning system is about 3 cm, the focal length f is 3.6 mm, pixel size μ is 2.8 μm, and the target distance D is 20 m, then a small Δx of 0.1 pixel will lead to a ΔD of 1 m. We can conclude that using the monocular camera alone without Leddar point clouds in this case cannot produce an accurate point cloud. Although it is feasible to integrate both Leddar depth and camera parallax in a bundle adjustment equation to reconstruct 3D points, in our ill-posed case, the camera has a highly limited contribution to recovering depth (Z) information. The main role of the camera in the bundle adjustment is to regularize the point clouds in the 2D plane (XY). After fine bundle adjustment, the camera can also be useful for filling gaps between the segments from the interpolation point of view. The interpolation methods are many (e.g., De Silva et al. [41] match the resolution between image and point clouds using a Gaussian process model for dense 3D recovery).

The objective of this study is to (1) configure a compact and low-cost fusion scanning system (FSS) including a multi-segment Leddar, monocular camera, and rotational robotics; and (2) propose an entire framework of calibration and fusion algorithms that produce dense colored point clouds covering a hemispherical view for 3D canopy monitoring. The specific technical refinements are (1) the addition of a kinematic motion constraint to the spatiotemporal bundle adjustment equations for the FSS calibration, and (2) the iterative optimization of monocular camera parallax and Leddar depth under ill-posed conditions. These technical contributions enable a cost-effective FSS, providing dense point clouds with rapid canopy information, such as gap fraction and LAI, useful for monitoring foliage biomass status and changes. Most parts of the framework are automatic and aimed to reduce potential manual intervention. Undoubtedly, the development of diverse lightweight sensors, particularly LiDAR, has been expanding the manner in which our environment can be sampled and digitized. It is time to tailor the tool selection to the application demand, instead of simply pursuing ultimate resolution. For example, as Table 1 indicates, in the canopy monitoring situation, with emphasis on sensor scalability and durability, the FSS can stand out as a flexible and balanced option among the four canopy sampling tools. We expect our work could promote more research attention to the lightweight scanning systems as a cost-effective option for 3D environmental mapping and monitoring.

## 2. Materials and Methods

### 2.1. Hardware Customization and Data Processing Framework

The Leddar optical FSS consists of two sensors: a 16-segment Leddar sensor (Leddar M16) and a web camera sensor in a 3D printed enclosure (13.3 × 9.1 × 4.1 cm). The two lightweight sensors (<300 g) sit on top of a tilting arm (DDT560 Direct Drive Tilt) and a panning base, as shown in Figure 1. Beside and beneath the pan-tilt arms are rotational servos, which drive the pan-tilt movement and determine the angular resolution and span of the rotation. Specifically, the tilting servo (Hitec HS-5485HB) is a standard digital servo that rotates between 0 and 118° with a highest resolution of 0.6°. The pan servo (Dynamixel MX-12W) can have 360° rotation with 0.08° resolution and can feedback rotation angles in real time. Specifications of Leddar, camera, and servos are provided in Table 2. Camera video and Leddar distance data are collected and stored by a Raspberry Pi, and servo rotations are manipulated by an Arduino Mega 2560 board. The detailed connections between sensors, pan-tilt robotics, and electronic controllers are shown in Figure 2.

To scan a wide field of view (e.g., hemispherical view), the pan-tilt system rotation follows a common raster scanning scheme: setting a fixed tilt angle for one horizontal scanline rotation and changing the tilt angle for the next. For each horizontal scanline, recordings from the Leddar, the camera, and the servo are stored asynchronously in separate files by the Raspberry Pi. The Leddar sensor outputs timestamp, segment distance, echo amplitude, and the echo quality index with an updating frequency of 100 s^−1^. The Raspberry Pi camera captures 720p video with a rate of 25 frames per second (FPS). The rotation angle from the pan servo is saved every 4°. The Leddar and pan angle readings are synchronized based on their millisecond-level timestamps tagged by the Raspberry Pi. The camera videos do not have timestamps, and their timing is inferred from motion detection.

Constructing the hardware of the FSS is technically straightforward. The system components are inexpensive and uncomplicated compared to most commercial LiDAR scanning systems, yet the fusion of different low-resolution data sources to generate dense point clouds is the primary challenge. The framework of our multi-source data fusion is illustrated in Figure 3, including: (1) mapping discrete Leddar distances onto individual video frames to create “3D pixels,” (2) aligning video frames globally to cover a panoramic field of view, and (3) adjusting frame alignment and extrapolating 3D point clouds based on the “3D pixels.”

### 2.2. Coordinate System Conversion for Calibration

Image frames decomposed from camera video do not have timestamps directly linkable to Leddar timestamps. It is necessary to match the camera motion with the Leddar motion and assign timestamps to camera frames. The start and stop of camera motion are detected by optical flows from neighboring frames: frames with average pixel velocity above a threshold of 0.6 pixels are considered moving. With start and stop timestamps, all the timestamps of the moving frames can be recovered based on linear interpolation.

The synchronization of camera and Leddar enables one-to-one mapping between Leddar distances and camera frames. We need to further locate Leddar footprints on the pixels of each video frame. This is solved through calibration of the FSS, or specifically, determining the unknown boresight parameters that convert the Leddar coordinate system to the camera coordinate system. Calibrating the FSS is only needed once, which enables the step of applying calibration transformation in the framework in Figure 3.

Our reference coordinate system (RCS) is a right-handed Cartesian coordinate system. It has the same units as the world coordinate system (WCS), with its origin at the camera optical center, the x axis along the transverse direction of the image plane, the y axis along the longitudinal direction of the image plane, and the z axis along the camera’s optical view direction. The target 3D coordinates ${\bar{P}}_{t}$ in RCS can be parametrized in Equation (2):(2)$${\bar{P}}_{t}\;=\;\left(\begin{array}{c} T\\ 0\\ 0 \end{array}\right)+\left(D_{it}+D_{b}\right)\left(\begin{array}{c} \cos\left(\theta_{i}\right)\sin\left(\psi_{i}\right)\\ \sin\left(\theta_{i}\right)\sin\left(\psi_{i}\right)\\ \cos\left(\psi_{i}\right) \end{array}\right)$$ where t is a specific time point, T the horizontal location of Leddar optical center in RCS, polar angles $\theta_{i}$ and $\psi_{i}$ the orientation of i ^th^ Leddar segment in RCS, $D_{it}$ the distance measurement of i ^th^ segment, $D_{b}$ the bias of distance measurement, and ${\bar{P}}_{t}$ the target 3D coordinates in RCS. We consider T,
$D_{b},$
$\theta_{i}$, and $\psi_{i}$ to be constant during pan-tilt rotation to represent no relative movement between the Leddar and camera. Assuming all the segments are equiangular, angle $\psi_{i}$ can be presented by an arithmetic sequence parametrized with $\psi_{0}$ and $\psi_{\Delta }$ (Equation (2)).

The real-world coordinates of target P can be converted from ${\bar{P}}_{t}$ in RCS through the camera extrinsic parameters in Equation (3):(3)$$P\;={\;R}_{t}{\bar{P}}_{t}+T_{t}$$ where $R_{t}$ is the rotation matrix, and $T_{t}$ the translation vector.${\;R}_{t}$ can be characterized by Euler angles ($\alpha_{t},\;\beta_{t},$ and $\gamma_{t}$) following z  −  y −x rotation order. Assuming the initial rotation matrix is $R_{0}$ or Euler angles ($\alpha_{0},\;\beta_{0},$ and $\gamma_{0}$), the temporal change of $R_{t}$ during a horizontal scan can be represented by rotation matrix $R_{w}$ in Equation (4):(4)$$R_{t}\;=\;R_{0}R_{w}$$
$$R_{w}\;=\left(\begin{array}{ccc} 1-2\left(u_{y}u_{y}+u_{z}u_{z}\right) & 2\left(u_{x}u_{y}-u_{w}u_{z}\right) & 2\left(u_{w}u_{y}+u_{x}u_{z}\right)\\ 2\left(u_{x}u_{y}+u_{w}u_{z}\right) & 1-2\left(u_{x}u_{x}+u_{z}u_{z}\right) & 2\left(u_{y}u_{z}-u_{w}u_{x}\right)\\ 2\left(u_{x}u_{z}-u_{w}u_{y}\right) & 2\left(u_{y}u_{z}-u_{w}u_{x}\right) & 1-2\left(u_{x}u_{x}+u_{y}u_{y}\right) \end{array}\right);$$
$$T_{t}\;=\;\left(\begin{array}{c} X_{ct}\\ Y_{ct}\\ Z_{ct} \end{array}\right)\;=\;\left(\begin{array}{c} X_{c0}+a_{1}\phi_{x}+a_{3}\phi_{z}-\left(a_{01}\phi_{x}+a_{03}\phi_{z}\right)\\ Y_{c0}+b_{1}\phi_{x}+b_{3}\phi_{z}-\left(b_{01}\phi_{x}+b_{03}\phi_{z}\right)\\ Z_{c0}+c_{1}\phi_{x}+c_{3}\phi_{z}-\left(c_{01}\phi_{x}+c_{03}\phi_{z}\right) \end{array}\right);$$
where
$$\left(\begin{array}{c} u_{x}\\ u_{y}\\ u_{z}\\ u_{w} \end{array}\right)\;=\;\left(\begin{array}{c} \sin\left(\phi_{\alpha }\right)\cos\left(\phi_{\beta }\right)\sin\left(\omega_{t}/2\right)\\ \cos\left(\phi_{\alpha }\right)\sin\left(\omega_{t}/2\right)\\ \sin\left(\phi_{\alpha }\right)\sin\left(\phi_{\beta }\right)\sin\left(\omega_{t}/2\right) \end{array}\right);\;$$
$$R_{t}\;=\;\left(\begin{array}{ccc} a_{1} & a_{2} & a_{3}\\ b_{1} & b_{2} & b_{3}\\ c_{1} & c_{2} & c_{3} \end{array}\right),R_{0}\;=\;\left(\begin{array}{ccc} a_{01} & a_{02} & a_{03}\\ b_{01} & b_{02} & b_{03}\\ c_{01} & c_{02} & c_{03} \end{array}\right).$$
where $\phi_{\alpha }$ and $\phi_{\beta }$ define the horizontal rotation axis and $w_{t}$ is the horizontal rotation angle measured by the servo. The camera optical center $T_{t}$ also slightly moves during the horizontal scan, whose temporal change can be parameterized by its initial position ($X_{c0},\;Y_{c0},$ and $Z_{c0}$) and rotation origin ($\phi_{x}$ and $\phi_{z}$) in Equation (4).

Solving the calibration Equations (2)–(4) requires measurement of P and $D_{it}$ from the same target point. However, since the exact Leddar point is invisible from the web camera, it is impossible to measure the exact 3D coordinates for P in the real world. Instead, we can reduce the requirement of the 3D P by finding a planar target with constant Z values ($Z_{0})$, and arbitrary X and Y values. Therefore, combining Equations (3) and (4), ${\bar{P}}_{t}$ and in RCS should satisfy a planar constraint in Equation (5):(5)$$\left(\begin{array}{ccc} c_{1} & c_{2} & c_{3} \end{array}\right){\bar{P}}_{t}+Z_{ct\;}=\;Z_{0}$$
where $Z_{ct}$ is the Z component of $T_{t}.$ With Equations (2) and (5), the only two required measurements are $D_{it}$ at multiple time points and $Z_{0}$ of the planar target.

The above four equations form a set of nonlinear calibration equations with nine unknown intrinsic terms (T,
$D_{b},$
$\theta_{0},$
$\psi_{0},$
$\psi_{\Delta },$
$\phi_{\alpha },$
$\phi_{\beta },$
$\phi_{x},$ and $\phi_{z})$ and six initial extrinsic terms ($X_{c0},\;Y_{c0},$
$Z_{c0},$
$\alpha_{0},\;\beta_{0},$ and $\gamma_{0})$. Our solution is iterative. The initial extrinsic terms $X_{c0},\;Y_{c0},$
$Z_{c0},$
$\alpha_{0},\;\beta_{0},$ and $\gamma_{0}$ are solved using least-square regression of camera collinearity equations (Equation (6)):(6)$$x_{t}\;=\;K\bar{P}\prime_{t}\;={\;\mathrm{KR}}_{t}^{T}\left(P^{\prime }-T_{t}\right)$$
$$K\;=\;\left(\begin{array}{ccc} \frac{f}{\mu } & 0 & x_{0}\\ 0 & \frac{f}{\mu } & y_{0}\\ 0 & 0 & 1 \end{array}\right)\;$$
given additional measurements of pixel coordinates $x_{t\;=\;0}$ and the corresponding world coordinates $P^{\prime }$ (details in Section 2.3). The $R_{t}^{T}$ in Equation (6) denotes transposed $R_{t}$ in Equation (3). For simplicity, the camera intrinsic parameters in Equation (6) are fixed, including camera focal length f, pixel size μ, half image width $x_{0}$, and half height $y_{0}$ in pixels. Lens distortion is not considered in this study. Combining Equations (4) and (6), both extrinsic terms ($X_{c0},\;Y_{c0},$
$Z_{c0},$
$\alpha_{0},\;\beta_{0},$ and $\gamma_{0})$ and intrinsic terms ($\phi_{\alpha },$
$\phi_{\beta },$
$\phi_{x},$ and $\phi_{z})$ can be inferred by non-linear least-square regression, and further substituted into Equations (2), (3), and (5) to finally estimate the Leddar’s intrinsic parameters (T,
$D_{b},$
$\theta_{0},$
$\psi_{0},$ and $\psi_{\Delta })$.

After knowing all intrinsic and extrinsic parameters, locating Leddar points on camera images is feasible using Equations (2) and (6). It is also possible to roughly estimate a Leddar point P given $D_{it}$ and $w_{t}$ from pan-tilt angles based on Equations (2)–(4). Instead of using pan-tilt angles, we can also rely on camera global alignment in the following section for more precise inference of $R_{t}$ and $T_{t}$, thus estimate P from Equation (2).

### 2.3. Calibration Experiment

This section presents an example of system calibration, with a flat wall ($Z_{0}\;=\;0)$ being our calibration target (Figure 4). The left bottom corner of the wall was defined as the WCS origin. The sensors scanned the wall with a fixed tilt angle of about 15° and continuous horizontal angle from 50° to 140° (2.4° per second). A total of 70 frames were subsampled from the video with equal intervals for calibration. On the front wall was a 16 × 9 grid from an optical projector, for the purpose of calibrating the camera extrinsic parameters in Equation (6). As mentioned, the calibration equation (Equation (6)) required measuring the WCS coordinates $P^{\prime }$ of each projected circle center and extracting the corresponding pixel coordinates $x_{t}$ from camera frames. We used a grid of circular targets instead of a chessboard pattern to be more robust with edge detection error during the automatic extraction of $x_{t}$ from camera images. Since the camera had a limited field of view of around 50°, not all circles appeared in the camera images. Therefore, an ID number was assigned to each circle to help link $x_{t}$ and $P^{\prime }$ automatically.

Assuming the projector lens had no distortion, we manually measured the $P^{\prime }s$ of the four corner circle centers on the wall and then applied bilinear interpolation to get $P^{\prime }s$ of all the 144 circle centers. The process of extracting $x_{t}$ was challenging because circle projection on the wall from the camera’s viewing perspective exhibited elliptical shapes. Extracting elliptical centers is more difficult than extracting circular centers. We adopted the characteristic number ellipse detector (CNED) of Jia et al. [42] to coarsely detect ellipse parameters (centers and axes). All the settings of the CNED were set as default except that the characteristic number on collinear points (CNL), the parameter of rejecting linearity, was set to 10.0 instead of 3.0. Due to thickness of ellipse edges on the wall, redundant ellipses could be detected using the CNED. The next steps averaged the center and axes’ parameters over all redundant ellipses; edges were thinned using a morphological operator; and a finely fit ellipse [43] within each ellipse region was defined by its axes’ parameters. The ID number within each ellipse region was identified by optical character recognition (OCR) in MATLAB, with the character set constrained to numbers 0–9. Using ellipse IDs greatly expedited the search for correspondence between $x_{t}$ and $P^{\prime }$ from 70 camera frames. The accuracy of OCR recognition using MATLAB was about 80%. Improper IDs were later corrected by voting from the nearest four IDs, and the final recognition accuracy was 100% among the 70 frames. Given the corresponding $x_{t}$ and $P^{\prime },$ the least-square Newton–Raphson iterative method [44,45,46] was applied to minimize residuals in Equations (2)–(6) following the aforementioned steps in Section 2.2. At the end of each iteration, the robust Huber’s function [47] was applied to reduce the effect of residual outliers. Initial estimates of parameters were also required by the nonlinear Newton–Raphson method, in which $\alpha_{0},\;\beta_{0},$ and $\gamma_{0}$ were roughly set as 0°, 40°, and 0°; $X_{c0},\;Y_{c0},$ and $Z_{c0}$ were manually measured as 1.80, 1.08, and 1.09 in meters; and T,
$D_{b},$
$\theta_{0},$
$\psi_{0},$
$\psi_{\Delta },$
$\phi_{\alpha },$
$\phi_{\beta },$
$\phi_{x},$ and $\phi_{z}$ were 0.03 m, −0.44 m, 180°, 90°, 2.5°, 0°, 0°, 0.06 m, and 0.00 m.

### 2.4. Fusion-Based Dense Point Cloud Recovery from FSS

Calibration is a preliminary requirement for fulfilling the framework in Figure 3. The framework focuses on generating and optimizing point clouds from multiple field scans. This section describes the suite of the visual odometry algorithms adopted, with a field experiment presented in the following section. Our field scans cover a hemispherical view with four scanlines spanning a vertical angle between 0° and 120°, though more scanlines can be added if desired. To avoid data overhead, we chose systematic sampling of the moving frames. Hence, our full hemispherical scan contains 150 × 4 moving frames with horizontal overlap of >90% and vertical overlap of ~80%. Aligning frames is the first problem, since the extrinsic parameters in the lab calibration environment are not repeatable in a new location. The intrinsic parameters, such as T,
$D_{b},$
$\theta_{0},$
$\psi_{0},$ and $\psi_{\Delta }$ from the Leddar sensor remain unchanged. To estimate the extrinsic parameters, target-based calibration or ground control points, while possible in some cases, would be tedious for a hemispherical view of 600 frames. Therefore, we directly used the dense and detailed photogrammetric information from the video to approximate camera poses $R_{t}$ and then incorporated Leddar distance into the bundle adjustment for fine camera extrinsic parameters ($R_{t}$ and $T_{t})$.

A common way of aligning multiple frames is to extract invariant features in each frame, match features between frames, and optimize the camera colinear equation (Equation (6)). Frames with multiple scanlines also need iterative correction of scanline skewness caused by the uneven distribution of matching features. The set of global alignments has been supported by image stitching software, such as PtGUI, for this study. A weakness of using off-the-shelf stitching software is the limited number of identified matched pixels, which is insufficient for bundle adjustment needs. Therefore, intensive extraction of SURF features [48] is added to the workflow in Figure 3, with outlier features filtered out using the homograph-based RANSAC algorithm [40]. The extracted SURF features are then matched between frames. Note that a 3D point can correspond to a set of matched pixels from multiple frames. The matched pixels from more than three frames are called “key pixels” here, whose features can be considered stable and will be used for the bundle adjustment later.

The global alignment uses PtGUI exports’ Euler angles of each frame, which can be used to interpolate Euler angles ($\alpha_{t},\;\beta_{t},\;\mathrm{and}\;\gamma_{t})$ or $R_{t}$ at any timepoint when Leddar distances are measured. Then Leddar point clouds $P_{t}$ can be roughly recovered using Equations (2) and (3), assuming $T_{t}$ is a zero vector. Projecting the Leddar point clouds $P_{t}$ back to each image frame will add depth information to a few pixels (or “3D” pixels here). The depth values of the “3D pixels” are essential to the inference the depth values of the “key pixels.” Our inference method is region-based interpolation: (1) filtering the foreground in each image using the k-means clustering method (k = 2); (2) segmenting images using statistical region merging algorithm (level = 8) [49]; and (3) implementing inverse distance weighting (IDW) interpolation for each key pixel within each region. Compared to global interpolation, using region-based interpolation better maintains sharp boundary lines between different image regions [50].

Based on the 3D “key pixels,” both transformation matrices $R_{t}$ and $T_{t}$, and WCS coordinates $P_{t}$ can be finely estimated using iterative bundle adjustment. First, $P_{t}$ estimated from 3D “key pixels” are averaged among different frames and reprojected to each image frame using Equation (7):(7)$$\left(\begin{array}{c} x_{t}\\ y_{t}\\ z_{t} \end{array}\right)\;={\;\mathrm{KR}}_{t}^{T}\left[\left(\begin{array}{c} X_{t}\\ Y_{t}\\ Z_{t} \end{array}\right)-T_{t}\right]$$
(8)$$\underset{\alpha_{t},\beta_{t},\gamma_{t},X_{\mathrm{ct}},Y_{\mathrm{ct}},Z_{\mathrm{ct}}}{\min}\left\|(\begin{array}{c} x_{t}^{\prime }\\ y_{t}^{\prime } \end{array}\right)-\left(\begin{array}{c} x_{t}/z_{t}\\ y_{t}/z_{t} \end{array})\right\|,\;\mathrm{using}\;\mathrm{nonlinear}\;\mathrm{regression}$$
(9)$$\left(\begin{array}{c} X_{t}^{\prime }\\ Y_{t}^{\prime }\\ Z_{t}^{\prime } \end{array}\right)\;=\;R_{t}K^{-1}\left(\begin{array}{c} x_{t}^{\prime }\\ y_{t}^{\prime }\\ 1 \end{array}\right)+T_{t}$$
(10)$$\underset{X_{t},Y_{t},Z_{t}}{\min}\left\|(\begin{array}{c} X_{t}\\ Y_{t}\\ Z_{t} \end{array}\right)-\left(\begin{array}{c} X_{t}^{\prime }\\ Y_{t}^{\prime }\\ Z_{t}^{\prime } \end{array})\right\|\;\mathrm{using}\;\mathrm{ridge}\;\mathrm{regression}\;\left(\lambda \;=\;0.01\right)$$

The reprojected points are normalized by Z coordinates and compared to the 2D coordinates of the “key pixels.” The least square error is minimized using nonlinear regression, with the camera’s extrinsic parameters ($\alpha_{t},$
$\beta_{t},$
$\gamma_{t},$
$X_{\mathrm{ct}},$
$Y_{\mathrm{ct}},$ and $Z_{\mathrm{ct}})$ as variables (Equation (8)). The new $P_{t}^{\prime }$ corresponding to “key pixels” is estimated using Equation (9) based on the optimized camera extrinsics. Note that $P_{t}$ is sensitive to small errors of “key pixels” and camera extrinsics due to the ill-posed mono-camera geometry. A robust solution to the ill-posed optimization problem is to use ridge regression [51], which partially minimizes the least square error between $P_{t}$ and $P_{t}^{\prime }$ in Equation (10). Its regularization parameter λ is set to be 0.01. The optimized $P_{t}$ from Equation (10) is again reprojected to each image frame in Equation (7), and repeat iterations from Equation (7) to Equation (10) until the error in Equation (8) is locally minimal. This iterative bundle adjustment for 3D recovery is similar to the gold standard method from Hartley and Zisserman [40], but with depth information provided from sparse Leddar distance instead of dense stereo geometry. The resulting point clouds will exhibit rich details in the 2D planar direction but limited variation in the depth direction.

The 3D recovery from bundle adjustment produces point clouds for “key pixels.” The “key pixels” are essentially from SURF feature extraction and are mostly focused on corner pixels with sharp color gradients. Other “internal” foreground pixels should also be incorporated to produce complete and dense point clouds. Our method is to extract dense foreground pixels and solve Equations (9) and (10) to create optimal dense 3D points. To satisfy Equation (10), one 3D point requires at least one pair of matching pixels from two frames. We already have matching pixels defined as “key pixels” from previous steps. We need to extrapolate the matching relationship for all foreground pixels. This is a pixel-level dense matching process. First, the foreground pixels need to be subsampled at a certain interval (e.g., 10 pixels in this study) to avoid data overhead. Then the disparities of “key pixels” between current frame and one matched frame are calculated. The disparities are used to interpolate a disparity map for all foreground pixels in the current frame. Given a disparity map, the foreground pixel location in the matched frame can be estimated. This dense matching step between a pair of matched frames is repeated for all the matching frames. Each set of matched pixels from multiple frames is given one unique ID, corresponding to one unique $P_{t}$. Finally, using Equations (9) and (10), the densely matched sets of pixels can produce dense point clouds. The final RGB colors of dense point clouds are the average RGBs from matched pixels.

### 2.5. Application: Tracking the Autumn Leaf Drop Processes with the FSS

The FSS was used to track canopy changes during an autumn season in 2018. Our experiment’s site was located in an area of cottonwood (poplar) stands in Lethbridge, Canada (49°41′45.2″ N, 112°51′54.0″ W). The FSS was mounted on a tripod surrounded by six poplar trees within 20 m, including *Populus angustifolia*, *P. deltoides* and their hybrid, *P. × acuminata* [52]. The irregular shapes of poplar trees increased the difficulty of depth information recovery but the rich texture of the scene facilitated feature extraction, which supports frame alignment. The camera lens filter was replaced to block near-infrared light and enable natural-colored images. Multiple scanlines were collected, each corresponding to 360° horizontal rotation at a speed of 2.4° per second. Four scanlines were selected to cover the upper hemispherical view of the scene for further processing. All scans were aligned, optimized, and densified to create colored point clouds using the methods in Section 2.4. The same scene was scanned with a Teledyne Optech ILRIS HD (1535 nm) TLS as a benchmark. The scanning angle was 360° × 80°, with the small zenith angle between 0°–10° not scanned. Only last returns were recorded and point spacing was 3.2 cm at a distance of 20 m from the TLS. A total of 30 ILRIS TLS scans were collected in 30 minutes, with three scanlines covering the entire upper hemispherical view. The TLS scans were co-registered by the iterative closest point (ICP) algorithm into one hemispherical scan with an average accuracy below 1.3 cm. The same scanning and processing activities were repeated on September 9th, September 17th, October 1st, and October 17th during the autumn defoliation period in 2018, to evaluate the reusability of our static scanning system in a temporal monitoring context. Hemispherical photos based the digital hemispherical photography (DHP) methods were also captured on September 9th, September 17th, and October 17th for benchmarking purposes.

Canopy vertical volume profile and plant area index (PAI) were extracted from the FSS point clouds to evaluate the capabilities of 3D canopy detection and canopy attribution. The volume profile was defined as a total volume of voxels at each height, where a unit voxel was 0.1 × 0.1 × 0.1 m and a height slice was 0.1 m. The PAI was calculated based on a path length distribution (PATH) model [7,15]. Specifically, the PATH model consisted of two equations (Equations (11) and (12)) with the gap fraction $\bar{P\left(\theta \right)}$ and path distribution $p_{l}$ as the only two inputs. To calculate angle-specific gap fractions, a hemispherical image from either FSS or TLS point clouds was first converted to a black and white binary image under a fisheye perspective. The fisheye binary image was equally sliced into 28 rings representing a zenith angle between 15°–69° and a ring width of 4°. The overlap between two neighboring rings was 2°. The gap fraction $\bar{P\left(\theta \right)}$ was defined as the ratio of the “hole” pixel numbers to the “filled” pixel numbers within a ring slice. The “filled” pixels represented the overall canopy area and was generated based on image morphological smoothing. The path distribution $p_{l}$ was defined as the probability density function (PDF) of the optical path length within the crown area, with l representing the within-crown path length. The l ranged between 0 and 1, scaled by the maximum value $l_{max}.$ The $p_{l}$ was approximated by the histogram of the l, normalized by the total histogram area. With both the gap fraction $\bar{P\left(\theta \right)}$ and path distribution $p_{l}$ extracted from crown area, the integral equation (Equation (11)) was solvable based on any root-finding algorithm, and the $FAVD\cdot l_{max}$ could be estimated; PAVD stands for the plant area volume density and G(θ) the leaf angle distribution. The G(θ) was set to 0.5 in this study, corresponding to a spherical leaf angle distribution [53]. The $PAVD\cdot l_{max}$ was then input to Equation (12) to determine the $PAI_{true}\left(\theta \right)$ at a specific zenith angle θ. The final PAI value was a weighted sum of $PAI_{true}\left(\theta \right)$ over all zenith angles (Equation (13)) [15].
(11)$$\bar{P\left(\theta \right)}\;=\;\int_{0}^{1}e^{-G\left(\theta \right)\cdot \left(FAVD\cdot l_{max}\right)\cdot l}p_{l}dl,\;\mathrm{where}\;\int_{0}^{1}p_{l}dl\;=\;1$$
(12)$$PAI_{true}\left(\theta \right)\;=\;\int_{0}^{1}\cos\left(\theta \right)\cdot \left(FAVD\cdot l_{max}\right)\cdot l\cdot p_{l}dl$$
(13)$$PAI_{true}\;=\;\frac{\sum_{\theta }PAI_{true}\left(\theta \right)\cdot \sin\left(\theta \right)}{\sum_{\theta }\sin\left(\theta \right)}$$

An important portion of the canopy was foliage and the corresponding index was LAI. The LAI was directly related to photosynthetic processes and carbon productivity, and was a more sensitive index than PAI to reflect seasonal biomass changes. We estimated LAI values by contrasting leaf-on and leaf-off gap fraction values, as illustrated in Equation (14), where N was the number of pixels and P was short for the gap fraction $\bar{P\left(\theta \right)}.$ Specifically, $N_{wood},$
$N_{leaf}$, and $N_{hull}$ are the numbers of wood, leaf, and canopy pixels, respectively, and $P_{off},$
$P_{on},$ and $P_{leaf}$ were the gap fractions of leaf-off, leaf-on, and leaf-only canopies, respectively. Based on Equation (14), $P_{leaf}$ was a simple ratio of $P_{off}$ to $P_{on}$ (Equation (15)). With known $P_{leaf},$ LAI was estimated in a similar manner with PAI using Equations (11)–(13), except for replacing $\bar{P\left(\theta \right)}$ (namely, $P_{on})$ with $P_{leaf}.$
(14)$$P_{off}\;=\;1-\frac{N_{wood}}{N_{hull}},\;P_{on}\;=\;1-\frac{N_{wood}+N_{leaf}}{N_{hull}},\;P_{leaf}\;=\;1-\frac{N_{leaf}}{N_{hull}-N_{wood}}$$
(15)$$P_{leaf}\;=\;\frac{P_{on}}{P_{off}}$$

## 3. Results and Discussion

### 3.1. Calibration

An example frame among the 70 frames from the camera video is shown in Figure 5a. The original 720p frame is cropped to 1280 × 500 to be compact. Only part of the 16 × 9 circle grid is within the field of view. The frame shows no blurry effect, implying that an FPS of 25 is sufficient to match a rotation speed of 2.4° per second. The purple band in the frame is the near-infrared LED light from Leddar, because the camera lens has no IR filter. The visibility of the LED light provides an intuitive way of validating calibration accuracy: the footprints of all the 16 Leddar segments after calibration should fall within the purple area. The ellipses on the wall were detected with the CNED method, shown in green in Figure 5b. It is clear that a few incomplete ellipses were skipped and many redundant ellipses were created. That is a preliminary step of approximating ellipse ranges and locations. Fine ellipses after edge thinning and geometrical fitting are shown in random colors in Figure 5c overlaid by the edge image. Edge noise is inevitable but has a limited impact on the ellipse fitting results. Each ellipse center is marked as a green cross. The integer number inside each ellipse is the circle ID predicted with the OCR and posterior voting method. The OCR recognition confidence is also placed under each ellipse ID as a decimal number. Ellipses with low OCR confidence are removed, such as numbers 142 and 80. The remaining 33 ellipses still satisfy the minimal requirement of having four control points for the camera collinearity equation. Note that all the 70 frames have been inspected to have four or more ellipses at the beginning. After calibration, point-based footprints of the Leddar segments were projected on the example frame. The Leddar points basically fall within the LED light zone, except for the first segment on the less illuminated area.

The point clouds after calibration of Leddar distances and poses are displayed in Figure 6a, with the X–Y projection approximating a planar shape and the X–Z projection a linear shape. The entirety of the point clouds contains 16 segments, each creating 70 points along the horizontal rotation direction. Several points overlap when the Leddar is still static at the beginning or the end. The projected trajectory of each segment on the X–Y plane is not a straight line, because Leddar has a fixed tilt angle of approximately 15° upwards. A few lines are not smooth, and their noise is not systematic for all lines, probably less likely due to the servo movement but rather the Leddar distance’s measurement instability. The standard error is 1.03 pixels for optimizing the camera collinearity equation (Equation (6)), 3.84 pixels for optimizing the temporal rotation equation (Equations (4) and (6)), and finally, 9.7 mm in WCS for solving the Leddar distance equation (Equations (2), (3), and (5)). Equation (4) constrains the rotation matrix to a fixed rotational axis. Without Equation (4), solving Equation (6) for each frame separately is still feasible and yields a standard error of 2.34 pixels. However, the retrieved camera extrinsic parameters, such as the camera center locations $T_{t}$, shown as the green dots in Figure 6b, lose physical meaning and present irregular movements. Therefore, applying Equation (4) accounts for the real camera movement, shown as the white arc points in Figure 6b, thus enables more reliable calibration parameters. The final estimates of intrinsic parameters T,
$D_{b},$
$\theta_{0},$
$\psi_{0},$
$\psi_{\Delta },$
$\phi_{\alpha },$
$\phi_{\beta },$
$\phi_{x},$
$\phi_{z}$ are 0.070 m, −0.428 m, 180.84°, 89.39°, 3.32°, −0.69°, 12.19°, 0.027 m, and −0.0006 m.

### 3.2. Fusion-Based Dense 3D Recovery

The success of point cloud recovery hinges, to a significant degree, on the quality of aligning video frames, because the camera poses determine the general form and structure of point clouds. Our video scans of poplar trees on four different dates are aligned based on automatic solutions provided from PtGUI, including image matching, feature extraction, feature matching, horizon correction, bundle adjustment, and image mosaicking. Example alignment results for the October 1st and October 17th videos are visualized as 360° × 120° spherical panoramas under equirectangular projection in Figure 7a,b. The trees displayed leaf-on conditions on October 1st and were defoliated completely on October 17th. Both scenes were centered on a railway viaduct, and the lower part of the panorama was discarded. No obvious alignment gap or inconsistencies were found from the two images. The processing results of the leaf-on scene are visualized in Figure 7c–h. Figure 7c shows alignment of separate images and their seamlines in PtGUI without mosaicking and color blending. The colors of individual tiles in Figure 7c differ from each other due to sunlight variation during scanning. Yet based on visual inspection, the alignment of tiles is seldom affected by the color difference, indicating strong robustness of PtGUI’s feature extraction algorithms. The alignment errors estimated from bundle adjustment in PtGUI are 4.5, 3.0, 2.5, 2.5, and 2.2 pixels for the scenes of September 9th, September 17th, October 1st, and October 17th, respectively. The relatively large error of the September 9th is due to various factors, such as windy conditions, the thick canopy, and a cloudy sky.

Leddar points were reprojected as the red crosses in Figure 7d and overlaid with the panorama view of the October 1st scene, after applying Leddar intrinsic parameters from the calibration, and camera extrinsic parameters from PtGUI alignment. The Leddar points capture the basic scene structure nearby, except for upper canopy, distant ground, and thin branches. The minimum, average, and maximum detection ranges of Leddar in this scene are 1.64, 6.45, and 14.17 m, respectively. The Leddar point clouds have obvious gaps between the segments and on the ground due to missing signals. This problem of Leddar data sparsity limits potential applications, such as tree surveying and object detection, unless photographic information is integrated. Therefore, the iterative bundle adjustment is applied at the point cloud level to minimize the disagreement between Leddar reprojected pixels, camera pixels, and camera extrinsic parameters. Iterations of the bundle adjustment error, measured in pixels, are plotted for the four scanning dates in Figure 8. The initial error of bundle adjustment can be more than 8 pixels but will converge to a level comparable to PtGUI alignment error. The final errors from bundle adjustment were 2.8, 1.9, 2.0, and 1.8 pixels for September 9th, September 17th, October 1st, and October 17th, respectively.

The fusion-based point clouds after image background removal, iterative bundle adjustment, and dense matching recovery were reprojected into two panorama images, as shown in Figure 7e–f. Figure 7e displays reprojected pixels with RGB colors, and Figure 7f is the corresponding depth image with nearer objects showing brighter colors. The reprojection from point clouds to a hemispherical-view image is not simply one point per pixel, considering the previous dense recovery process has a subsampling rate of 10 pixels per point. Therefore, each point had a buffer of 10 pixels in a hemispherical image. Similarly, reprojecting TLS point clouds into a hemispherical depth image in Figure 7h also needs to consider the footprint of each TLS laser beam. The scanning spacing of each Ilris HD beam (1600 μrad) was set to be the footprint size according to the suggestions in [12]. The scanning spacing corresponds to a constant pixel size of 1.6, thus each reprojected pixel of the hemispherical images was dilated by a factor of 1.6.

In contrast to the Leddar reprojection image in Figure 7d, the image in Figure 7e not only captures rich 2D details but also covers a reasonable extent due to the region-based interpolation. The main problem of the fusion-based point clouds is false interpolation. The problem can be illustrated when comparing specific tree point clouds extracted from TLS, Leddar point clouds, and fusion-based point clouds in Figure 9a–c. The TLS point clouds clearly exhibit branch-level details with warmer colors representing higher laser intensity. The fusion-based point clouds have distinguishable stem colors and noisy branches, still highly detailed compared to the obscure Leddar point clouds. Yet the fusion-based point clouds overfill the gaps between branches and also falsely incorporate pixels from remote shrubs. This is inevitable because region-based interpolation and bundle adjustment can mitigate but not eradicate the problem of coarse and sparse depth measurement from Leddar. The depth image in Figure 7f displays a strong smoothing effect compared to the depth image reprojected from TLS in Figure 7h, but is much more detailed than the Leddar-only point clouds in Figure 7g with indiscernible sparse points.

### 3.3. Tracking Changes of Canopy Vertical Volume Profile, PAI, and LAI

Vertical volume profiles from TLS, fusion-based point clouds, and Leddar point clouds can be compared in Figure 10a–d. The *r*^2^ of profiles over the maximum height range between Leddar and TLS was noted for each date. The *p*-value based on a paired t-test between profiles is also provided. It is a rule of thumb to consider that two profiles have different mean values if the *p*-value is below a significance level of 0.05. Regardless of scanning date, both the profiles from the fusion-based point clouds and the Leddar point clouds were correlated with the TLS profiles. The *r*^2^ between the Leddar and TLS profiles remains at approximately 0.3 from the first three leaf-on scenes and increases to 0.48 for the last leaf-off scene. In contrast, the *r*^2^ of profiles between the fusion-based and TLS are around 0.65 from the leaf-on scenes, constantly higher than 0.52 from the leaf-off scene. The *r*^2^ improvement of fusion-based over Leddar is because thick crowns and leafy understory lead to Leddar signal loss but do not affect photography-based interpolation. The profile difference between the fusion-based and the reference TLS is mainly the higher frequency in the middle crown area due to the overfilling effect, and also thinner volume near the upper crown associated with the loss of supporting points from Leddar. For the first two leaf-on scenes, the overestimation effect near the middle crown is dominant, thus causing an obvious bias of mean values denoted by a low *p*-value of 0.000. For the last two scenes with increasingly defoliated crowns, the fusion-based point clouds tend to incorporate fewer false pixels from areas beyond the canopy, resulting in the retreat of the lower canopy. The fusion-based point clouds’ points are not as rich in the depth direction compared to TLS, thus the lower canopy parts of the last two scenes are thinner than TLS. As a result, the mean bias of profile is offset, and high *p*-values (>0.3) are found for the last two scenes. Note that the TLS has a slightly narrower scanning view than the FSS, with part of the upper crown and ground not sampled by scans. The profile difference around the upper crown and ground can be higher than observed in Figure 10. This problem of profile distortion might be due to the imperfect hemispherical stitching process.

The benefits of synthesizing both 3D and color information make FSS a potentially valuable complement to conventional LAI or PAI surveying tools, such as digital hemispherical photography (DHP). Figure 11 compares the fisheye image from FSS with the DHP photo from the same site. The canopy shapes in the two images are visually identical, seen in Figure 11a,b. The FSS, in addition, captures depth information shown in the image in Figure 11c, with a benchmarking TLS depth image provided in Figure 11d. Note that the upper crown area was not scanned with the TLS due to the field of view constraint. The availability of depth images enables FSS to calculate true PAI and LAI based on the PATH model. Different PAI and LAI estimates based on FSS, TLS, and DHP methods, and based on PATH and non-PATH methods, are contrasted in Figure 12, with bars denoting PAI and crosses indicating LAI. The non-PATH method relies on the LAI models from the Hemisfer software [54,55], which combines the leaf angle distribution (LAD) function of Lang [56]; the clumping correction of Lang and Xiang [57]; and the non-linearity correction model of Schleppi, Conedera, Sedivy, and Thimonier [54]. The non-PATH method focuses on the RGB images from DHP or FSS, or the depth images from TLS, whereas the PATH method additionally needs point cloud input from FSS or TLS.

For the non-PATH methods in Figure 12, PAI and LAI values generally decline with the defoliation dates, with all the LAI values reaching zero level on the leaf-off date October 10th, except that the PAI and LAI value from non-PATH FSS increases on October 1st. The incorrect increase implies the instability of using image-only methods. The possible cause of the incorrect increase is FSS’s underestimation of PAI and LAI from the September 9th and October 1st FSS images, in contrast to the PAI and LAI values from DHP and TLS. Strong spectral reflectance from sunlight is observed in the September 9th and October 1st FSS images and a small portion of canopy pixels in the FSS image displays a similar color as the sky background. These bright canopy pixels were not successfully identified as leaf area, causing the underestimation effect. The DHP method does not have the underestimation issue because the DHP images were captured near dusk. The TLS method does not have the stability issue of FSS, due to the fact that depth images are used instead of color images. The TLS method, however, has an issue of overestimating PAI. The leaf-off PAI from TLS is 32% higher than from DHP, compared to the average 3% overestimation of leaf-on PAI from TLS. The overestimation issue of TLS has two typical causes. The depth images, particularly the leaf-off ones, contain ghost points or misaligned points around thin branches and twigs. Gaps smaller than the beam width of TLS are also not differentiable from the depth images [12]. The overestimation by TLS of twig PAI leads to an underestimation of LAI by 26% relative to DHP.

With the PATH model applies to TLS and FSS, the PAI estimates are approximately 30%–45% higher than the non-PATH PAIs. The PATH PAI estimation from FSS does not have the problem of PAI increase on October 1st, indicating the importance of incorporating depth correction. The PATH LAI estimates are also higher than the non-PATH by 14% on average, except for the October 1st FSS LAI anomaly. Considering that the optical image methods usually underestimate true PAI or LAI [7] by 20%–60%, it is assumed that the PATH model is a closer approximation of the true PAI or LAI values.

It is important to understand why the PATH model usually outputs higher PAI (or LAI) values than the classic geometrical-optical model. Indeed, the $PAI_{true}\left(\theta \right)$ solved from the PATH model does not have a simple analytic form, due to various forms of $p_{l}.$ However, if we simply assume that $p_{l}$ is constantly 1, or equivalently, the within-crown path length distribution is uniform, the PATH model then has an analytical solution of PAI, which is basically a Lambert W function of gap fraction $\bar{P\left(\theta \right)}$ (Figure 13). The traditional LAI model (effective LAI) using Beer’s law [14,58] is also contrasted in Figure 13. It clearly shows that the PATH PAI is consistently higher than the non-PATH PAI, especially when the gap fraction is small. It is also noteworthy that the PATH model might be overly sensitive to near-zero gap fraction changes. The upper bound of PAI based on the PATH model is $-\frac{\cos\left(\theta \right)}{G\left(\theta \right)}\frac{l_{max}}{l_{min}}ln\bar{P\left(\theta \right)}$ and the lower bound is $-\frac{\cos\left(\theta \right)}{G\left(\theta \right)}\frac{l_{min}}{l_{max}}ln\bar{P\left(\theta \right)}.$ The wide range of PAI indicates strong flexibility of the PATH model, but it is also important for future studies to examine what the rigorous PAI bounds based on different forms of the $p_{l}$ functions are, and what a suitable analytical form of $p_{l}$ functions or PAI functions with a smoother gap fraction sensitivity can be. In addition, the PATH model is essentially a variant of Beer’s law. It does not account for the effect of laser footprint increase and density attenuation with distance. Integrating these laser-dependent factors to the PATH model is feasible, but solving the integral equations would become difficult. An alternative approach provided by [59] is to discard Beer’s law and model the statistical form, relating leaf area distribution, laser path length, density attenuation, and footprint size. Such a statistical model is solvable with a maximum likelihood estimator (MLE).

## 4. Conclusions and Future Work

Timely monitoring of canopy characteristics is necessary to understand the spatiotemporal variation of biomass in a forest ecosystem and to evaluate carbon budgets as part of forest stand reporting. The advent of low-cost multi-segment LiDAR sensors, Leddar in particular, has presented many successful object tracking applications. Yet the Leddar sensor is not comparable to TLS in sampling 3D details due to its limited FOV and point resolution. This limitation was mitigated in this study by constructing a low-cost 3D fusion scanning system, FSS, integrating Leddar, camera, and pan-tilt robotics. A framework of integration was developed, generally including (1) plane-based physical calibration, converting Leddar distance into a 3D point and locating the Leddar point from images; (2) global image alignment, obtaining panorama and coarse camera poses; (3) iterative bundle adjustment, optimizing camera poses using both Leddar distance and corresponding pixels at the point cloud level; and (4) dense point cloud recovery, based on density matching and interpolation. The calibration error of Leddar points was 9.7 mm at a distance of ~1 m. The set of fusion-based methods was applied to recover hemispherical colored point clouds from multi-temporal poplar tree scans during the autumn defoliation period. The bundle adjustment error was 1–3 pixels, indicating a strong agreement between image and Leddar projection on the X–Y plane. However, uncertainty existed in the depth (Z) direction due to the coarse resolution of Leddar distance. Final fusion-based point clouds were compared to the TLS scans collected on the same spot and date. The vertical profile volumes between TLS and FSS point clouds had an *r*^2^ of 0.5–0.7 over the maximum tree height range, which varied with leaf cover conditions and exceeded the *r*^2^ of 0.3–0.5 between TLS and pure Leddar point clouds. PAI and LAI metrics were also extracted from FSS, TLS, and DHP for leaf-on and -off dates. Using only image data, the PAI and LAI tended to be underestimated with FSS and overestimated with TLS. With the point cloud PATH model, both PAI and LAI from FSS or TLS were corrected to approach their assumed true values. By combining both color and depth information, the FSS has demonstrated versatility and significant potential for the application of canopy foliage monitoring.

The FSS system was primarily developed for static scanning of the environment. For environmental applications, such as crown measurement and biomass delineation, low-cost sensor systems such as the FSS (as built) do not parallel the resolution or precision of TLS and DHP at present, yet the demand for gross mensuration should not be overlooked, and sensor hardware upgrading is inevitable. The core contribution of this study is a holistic calibrating and fusing scheme for a low-resolution multi-sensor platform. The advantage of FSS should be clear. It is more portable and of a lower cost compared to a conventional TLS, and has a higher frequency, and improved detection range and FOV compared to many indoor-oriented LED or flash LiDAR systems. It is, therefore, suitable to be deployed in high numbers into sensor networks for broad-scale environmental monitoring, which will be our future work. Adapting the FSS to other mobile systems such as UAV is also feasible, following a similar framework of calibration, pose approximation, bundle adjustment, and densification, except for the need for external pose information from GPS and IMU, and a dedicated image alignment method. A better viewing geometry, such as stereoscopy from a mobile system, could substantially improve the precision of 3D recovery, particularly in the depth direction. In addition, the fusion scheme presented in this study focuses on the data aggregation level, whereas a higher level of fusion based on spatiotemporal attributes, patterns, and mission management still requires investigation. A bright future for cost-effective, fusion-based 3D canopy monitoring systems is anticipated.

## Figures and Tables

**Figure 1 sensors-19-03943-f001:**
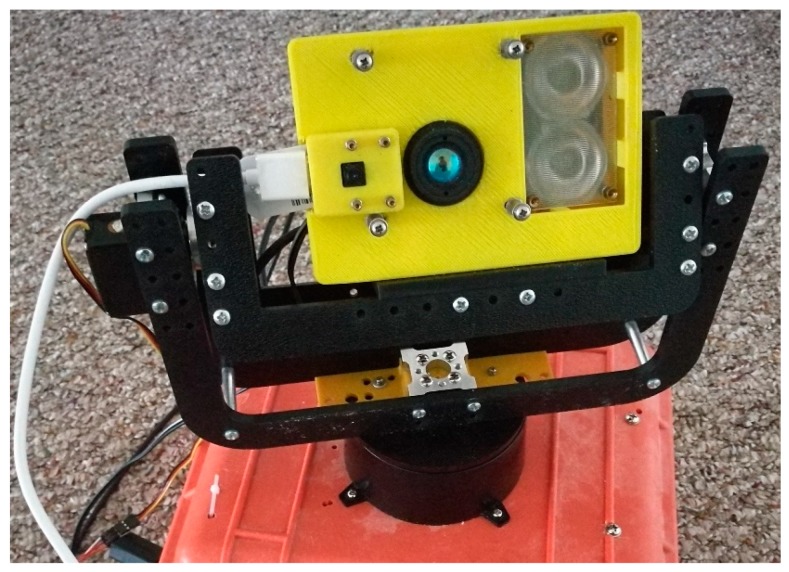
A fusion scanning system (FSS) with light emitting diode detection and ranging (Leddar) and monocular camera sensors.

**Figure 2 sensors-19-03943-f002:**
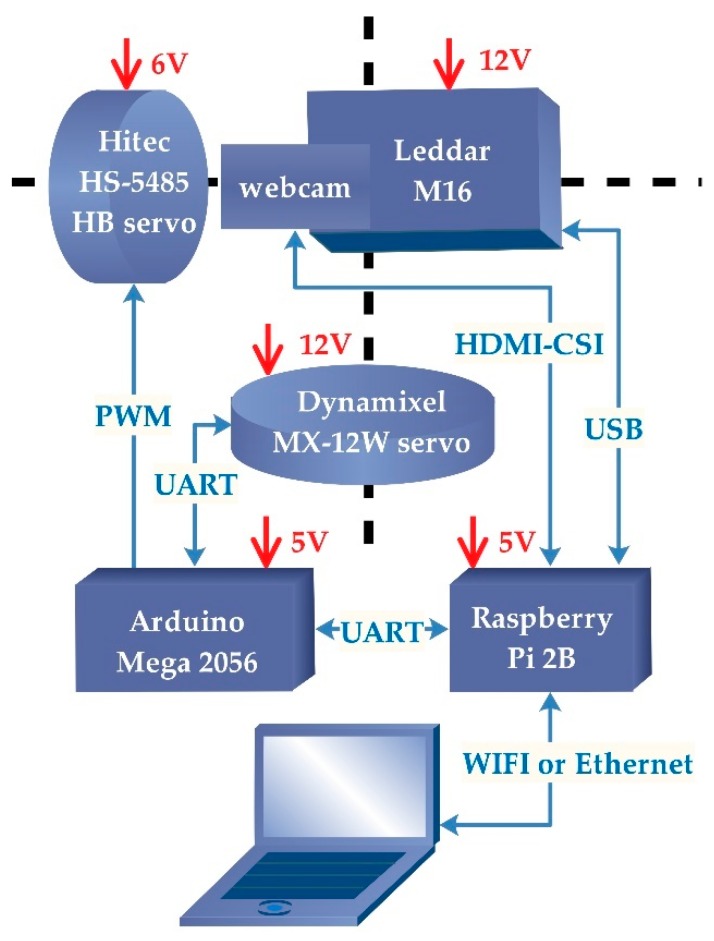
Hardware components and connections of the FSS. PWM: Pulse width modulation. UART: Universal asynchronous receiver/transmitter. CSI: Camera serial interface.

**Figure 3 sensors-19-03943-f003:**
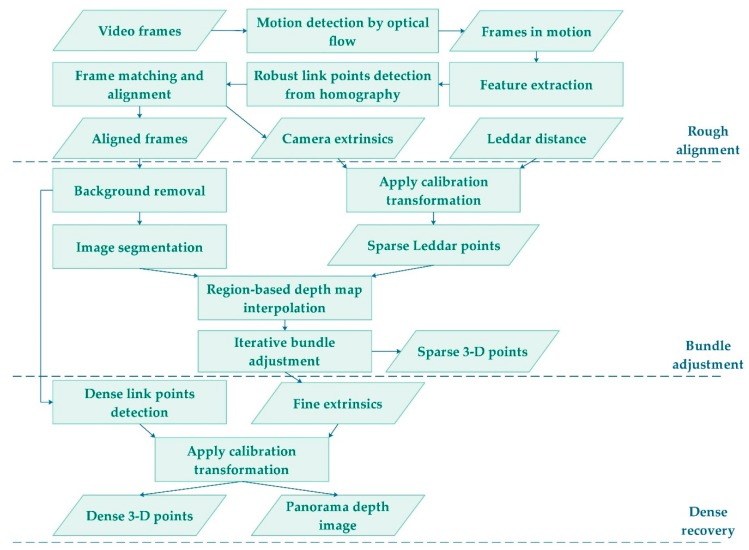
Framework of point cloud recovery from monocular camera and sparse Leddar segments.

**Figure 4 sensors-19-03943-f004:**
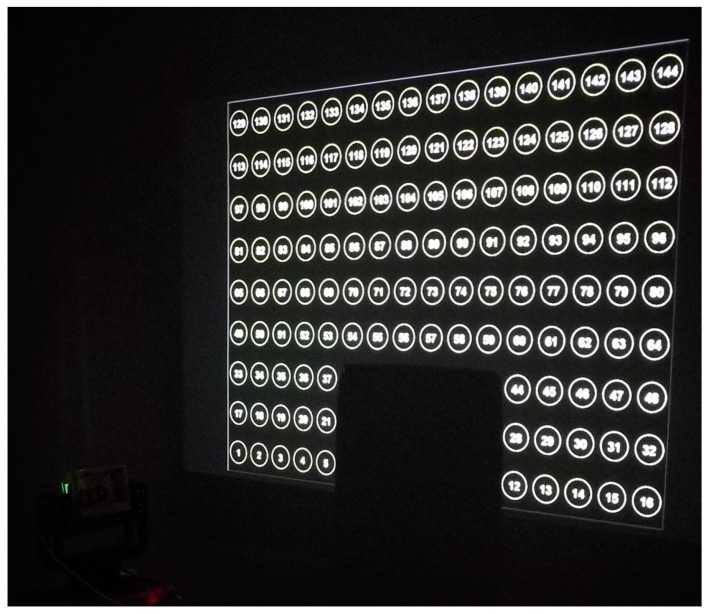
Experiment setup for the FSS calibration.

**Figure 5 sensors-19-03943-f005:**
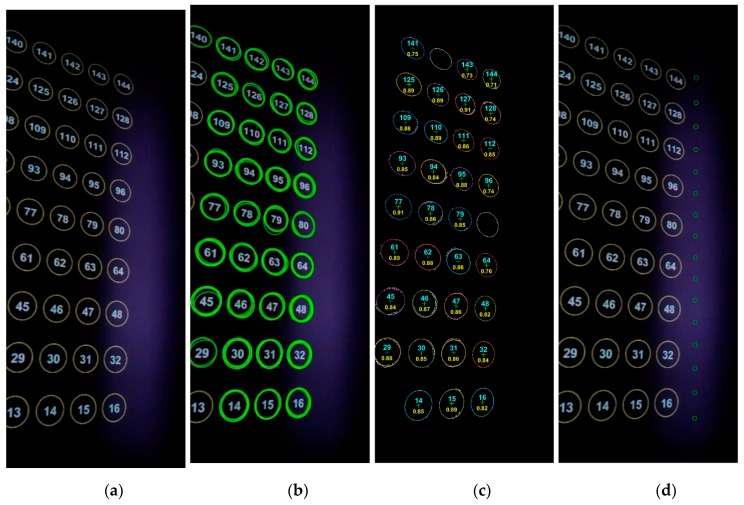
Calibration processing of an example frame: (**a**) circle grid (yellow) and LED light (purple) from camera view, (**b**) ellipse detection (green) by CNED method, (**c**) fine ellipse fitting (random color), ellipse ID from OCR (cyan) and OCR confidence (yellow), and (**d**) calibrated segment points reprojected to the frame image (green).

**Figure 6 sensors-19-03943-f006:**
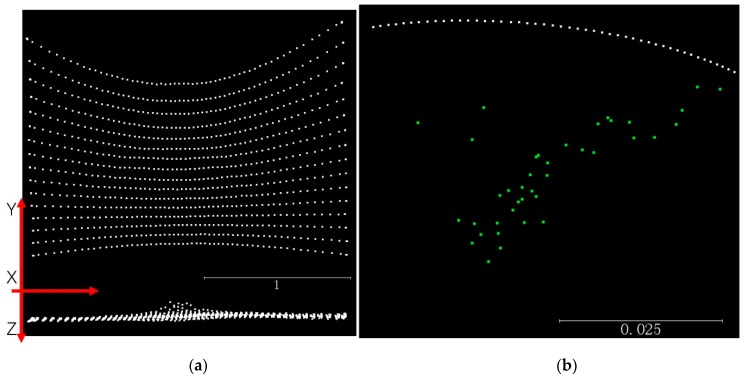
Calibrated Leddar points and camera trajectory: (**a**) calibrated Leddar points of a flat wall on the X–Y plane (above) and on X–Z plane (below), and (**b**) camera trajectory with rotational constraints (above) and without constraints (green below).

**Figure 7 sensors-19-03943-f007:**
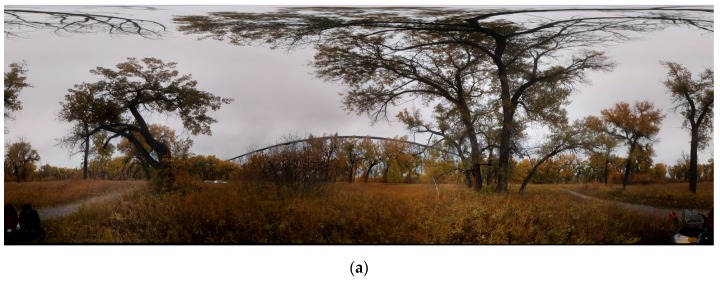
Hemispherical view of processing results: (**a**) image global alignment for the October 1st scan, (**b**) image global alignment for the October 17th scan, (**c**) global alignment layout in PtGUI software with image IDs and seamlines for the October 1st scan, (**d**) Leddar-only point clouds (red cross) reprojected to the hemispherical image, (**e**) RGB colors from fusion-based point clouds, (**f**) depth image from fusion-based RGB point clouds, (**g**) depth image from Leddar-only point clouds (point size enlarged for clearer visualization), and (**h**) the depth image from TLS scans. (**e**–**h**) were all done using hemispherical projection.

**Figure 8 sensors-19-03943-f008:**
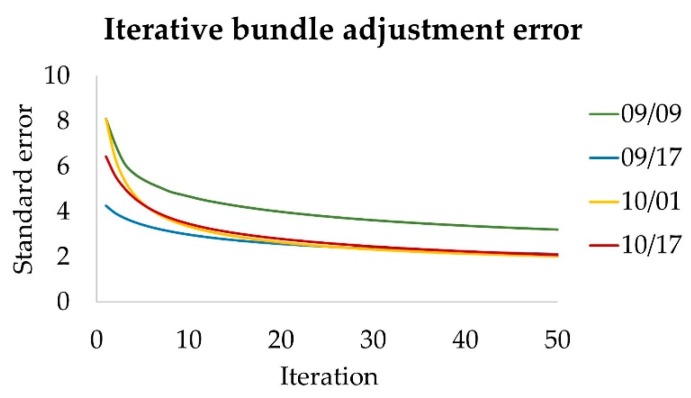
Fusion error convergence with iterations (in pixels) on four defoliating dates in 2018.

**Figure 9 sensors-19-03943-f009:**
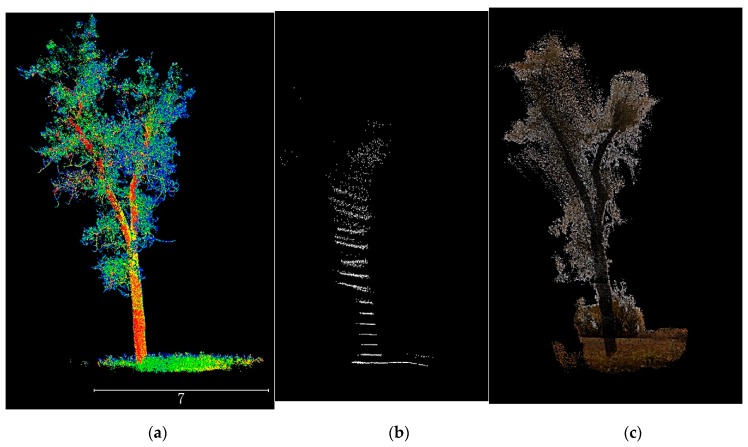
Example tree point clouds from (**a**) TLS scans, (**b**) Leddar-only point clouds, and (**c**) fusion-based dense point clouds.

**Figure 10 sensors-19-03943-f010:**
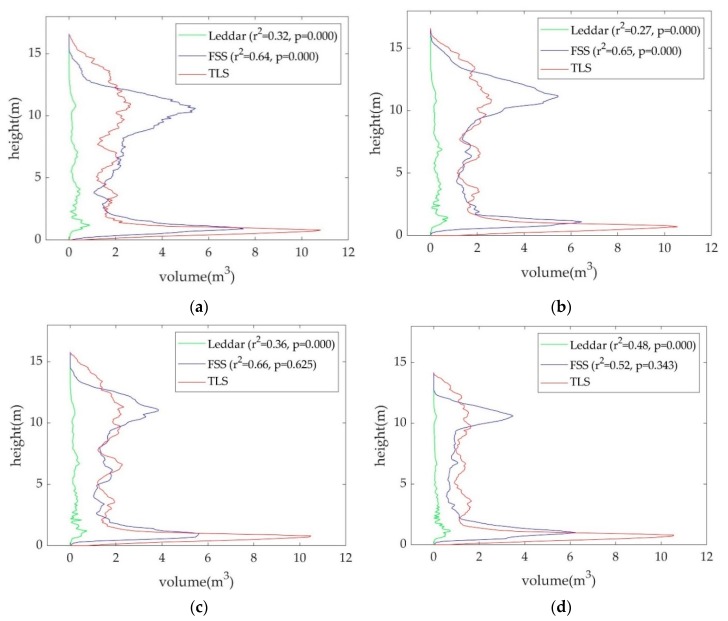
Vertical volume profiles from TLS, fusion-based, and Leddar-only point clouds on (**a**) September 9th, (**b**) September 17th, (**c**) October 1st, and (**d**) October 17th, 2018, where the horizontal axis denotes volume of voxels with a unit voxel of 0.1 m^3^, and the vertical axis denotes height in meters.

**Figure 11 sensors-19-03943-f011:**
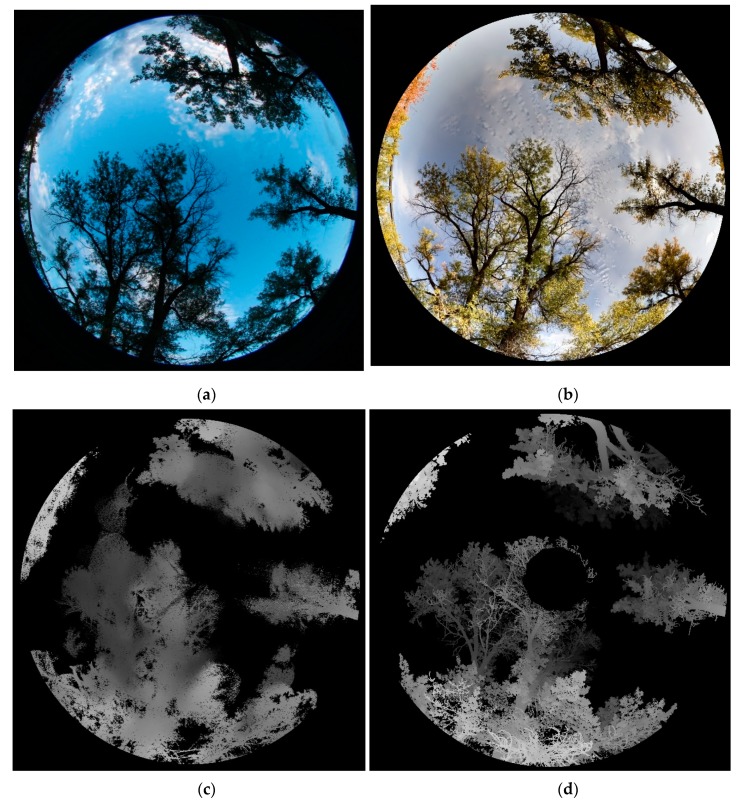
Fisheye-view images compiled from the September 9th datasets based on (**a**) DHP, (**b**) FSS, (**c**) FSS depth, and (**d**) TLS depth.

**Figure 12 sensors-19-03943-f012:**
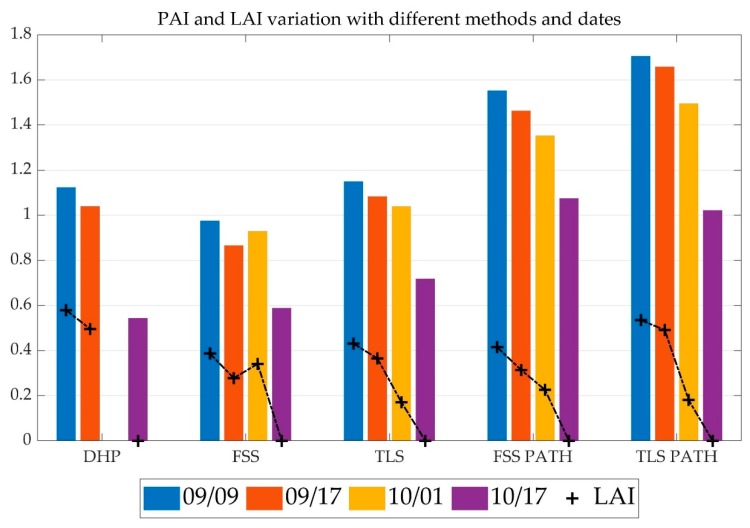
Comparing different methods of leaf area index (LAI) estimation on four scanning dates. The colored bars represent plant area index (PAI), and crosses for the associated LAI. The October 1st DHP dataset is not available. The DHP and the FSS methods are based on RGB images and the TLS based on depth images.

**Figure 13 sensors-19-03943-f013:**
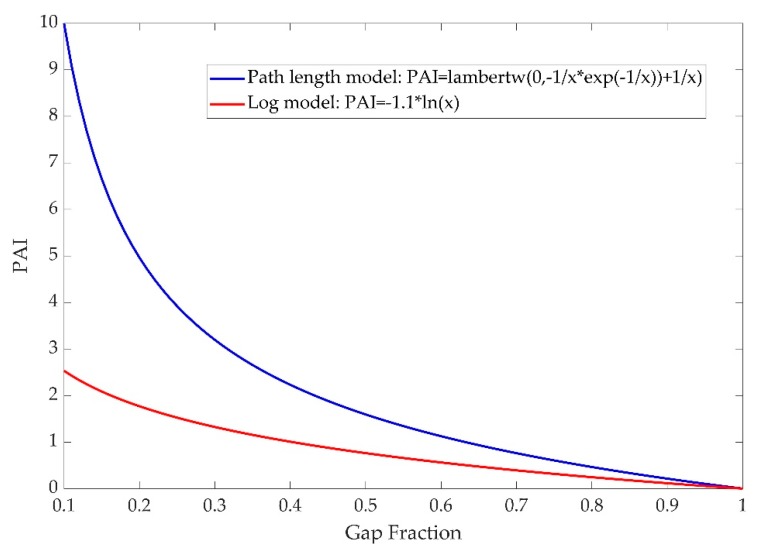
Relationship between gap fraction and PAI (or LAI). The red curve shows PAI values from the path length distribution model, compared to the blue curve from the simple Beer’s law model. Detailed mathematic functions are provided in the legend, with *x* representing the gap fraction.

**Table 1 sensors-19-03943-t001:** Qualitative ranking of advantages among terrestrial laser scanning (TLS), digital hemispherical photography (DHP), fusion scanning system (FSS), and sweeping 2D LiDAR for canopy sampling. Ranks are denoted as ++, + and - in descending order.

	TLS	DHP	FSS	2D LiDAR
Spatial resolution	++	++	+	-
Detection range	++	+	+	-
Equipment affordability	-	-	+	++
Operative efficiency	+	++	+	+
3D measurement accuracy	++	-	+	+
Portability and scalability	-	-	+	++
Repeatability and durability	-	-	+	+

**Table 2 sensors-19-03943-t002:** Major hardware specifications.

Camera	Leddar	Tilt Servo	Pan Servo
OmniVision OV5647	M16 module	Hitec HS-5485HB	Dynamixel MX-12W
FOV: 54° × 41°	Distance: 0 to 50 m	Max angle: 118°	Max angle: 360°
Lens: f = 3.6 mm, f/2.9	Frequency: ≤100 s^−1^	PWM: 750–2250 μs	Steps: 4096
Calibration: no IR	Wavelength: 940 nm	Deadband: 8 µs	Resolution: 0.088°
Application: IR filter	Power: 12/24 V, 4 W	Power: 4.8–6.0 V	Voltage: 12 V

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
