# Peer review of "A Lightweight Leddar Optical Fusion Scanning System (FSS) for Canopy Foliage Monitoring"

_sensors, 2019, doi:10.3390/s19183943_

Round 1

Reviewer 1 Report

It is recommended to mention clearly the qualitative advantages of FSS in the proposed application area.

Author Response

Your comments are greatly appreciated. Please see all our responses in the attached document.

Reviewer 2 Report

This article is well written. It explains well about the sensor calibration and data fusion.

High resolution images should be used for Figures 2 and 3. The current ones look blurred when printed.

The r^2 scores of volume differences between TLS and Leddar-only point clouds are small, and therefore paired t-tests over them would be obvious. But the r^2 scores between TLS and FSS point clouds are high, and it would be interesting to see the results of paired t-tests over them.

Author Response

(The authors gave the same response as above.)

Reviewer 3 Report

The paper describes low cost scanning system for canopy monitoring.  While the paper is well-written and might be interesting for the readers, following are some issues that I found it:

While the paper is technically sound, it is not clear what the actual scientific contributions of the paper are. There are large sections of the paper where it is not clear what the authors’ work is and what did the authors picked up from the existing literature. As for example, Section 2.2 gives a short overview of well-known transformations between the coordinate systems that is presented as the authors’ novelty. It is not clear to me why the authors address their system as data fusion system. In my opinion, it deals only with data aggregation or, in best case, data fusion at level 0 of e.g. JDL/DFIG fusion model.

Following are some additional comments:

L. 22-23: Scientific papers are not the place for such speculations.

L. 118: There is no need for Eq.1 as these information are beyond the scope of motivation / related work.

L.129: Chapter?

L.130 – 136: The objective 2 is confusing and, in general, I do not see any need for the information written here.

L.137: Again, scientific paper is not the place for speculations.

L. 593: This is the conclusion section and, thus, the conclusions of the study should be presented here. The abstract of the study should be explained in the abstract.

Author Response

(The authors gave the same response as above.)

Round 2

Reviewer 3 Report

I still believe the paper is week from scientific perspective, however, due to its potentially interesting content for the readers, I believe it can be published as it is.

Author Response

The emphasis of this technical note is a proof of concept that the fresh LED-based LiDAR technology is adaptable to the foliage scanning purpose. Exploring a deeper scientific core in terms of the sensor design, system integration and environmental application is desired and definitely would be our next goal. Thank you for providing all the valuable comments.